



**The contributions of the leading modes of the North Pacific sea surface temperature**
**variability to the Arctic sea ice depletion in recent decades**
Lejiang Yu[1], Shiyuan Zhong[2], Timo Vihma[3]
1 State Oceanic Administration Key Laboratory for Polar Science, Polar Research Institute of
China, Shanghai, China
2 Department of Geography, Environment and Spatial Sciences, Michigan State University, East
Lansing, MI, USA
3 Finnish Meteorological Institute, Helsinki, Finland
*Corresponding Author's address Dr. Lejiang Yu
SOA Key Laboratory for Polar Science, Polar Research Institute of China, Shanghai, China
Jinqiao Road 451, 200136, Shanghai, China
Phone: 0086-020-58712034, email: yulejiang@sina.com.cn



**Abstract.** Arctic sea ice decrease in extent in recent decades has been linked to sea surface
temperature (SST) anomalies in the North Pacific Ocean. In this study, we assess the relative
contributions of the two leading modes in the North Pacific SST anomalies representing external
forcing related to global warming and internal forcing related to Pacific Decadal Oscillation (PDO)
to the Arctic sea ice loss in boreal summer and autumn. For the 1979-2017 period, the time series
of the global warming and PDO modes show significant positive and negative trends, respectively.
The global warming mode accounts for 44.9% and 50.1% of the Arctic sea ice loss in boreal
summer and autumn during this period, compared to the 20.0% and 22.2% from the PDO mode.
There is also a seasonal difference in the response of atmospheric circulations to the two modes.
The PDO mode excites a wavetrain from the North Pacific to the Arctic; the wavetrain is not seen
in the response of atmospheric circulation to the global warming mode. Both dynamic and
thermodynamic forcings work in the relationship of atmospheric circulation and sea ice anomalies.














## 1 Introduction


Accompanying the abrupt Arctic warming, Arctic sea ice has exhibited a sharp decline trend
in recent decades. To explain the Arctic sea ice loss, researchers have proposed a variety of
feedback mechanisms, including ice-albedo feedback (Flanner et al., 2011), water vapor and
cloud-radiative feedback (Sedlar et al., 2011), and atmospheric lapse-rate feedback (Bintanja et al.
2011; Pithan and Mauritsen, 2014). These feedback mechanisms exert effects on Arctic sea ice in
the context of the changes in both the anthropogenic forcing and the large-scale circulations. In
this study, we assess the impacts of these two factors on Arctic sea ice loss.
The anthropogenic factor mainly includes greenhouse gas and aerosol emissions. The
increase in greenhouse gas concentrations and the overall decrease in aerosol emissions have been
linked to the observed Arctic sea ice loss (Min et al., 2008; Notz and Marotzke, 2012; Gagné et al.
2015). The natural factor, mainly changes of large-scale atmospheric and oceanic circulations, has
also contributed to the Arctic sea ice decline. The decrease in Arctic sea ice extent has been linked
to a positive trend in the North Atlantic Oscillation (NAO) (Deser et al., 2000), the Arctic
Oscillation (AO) (Rigor et al., 2002) and the Arctic Dipole (AD) (Wang et al. 2009) indices. The
multidecadal variability of sea surface temperature (SST) in the North Pacific and Atlantic Oceans
referred to as the Pacific Decadal Oscillation (PDO, Mantua et al., 1997) and the Atlantic
Multidecadal oscillation (AMO, Enfield, 2001) also have a strong influence on Arctic sea ice by
affecting atmospheric circulation and oceanic heat transfer (Woodgate et al., 2012; Ding et al.,
2014; Yu et al., 2017; 2019; Zhang, 2015).
It is difficult to separate the contributions of natural (internal) and anthropogenic (external)
forcings to the Arctic sea ice decline. General circulation models (GCM) have been applied to



assess the relative contributions of these forcings and GCM simulations have suggested a
contribution from internal forcing ranging from 20% to 50% over the last three decades (Stroeve
et al., 2007; Kay et al., 2011; Day et al., 2012; Ding et al., 2019). However, results from GCMs
have been found to underestimate the observed Arctic sea ice loss (Winton, 2011; Stroeve et al.,
2012; Mahlstein and Knutti, 2012) due possibly to low sea ice sensitivity to greenhouse gas
emissions (Notz and Stroeve, 2016; Rosenblum and Eisenman, 2017) and internal climate
variability (Kay et al., 2011; Stroeve et al., 2012; Notz, 2014; Swart et al., 2015).

A recent study noted a close connection between the Arctic sea ice loss and the changes in

SST in the North Pacific Ocean (Yu and Zhong, 2018) in recent decades. The main modes of
variability in the North Pacific SST include the global warming mode, PDO mode (Wills et al.,
2018) and Victoria mode (Bond et al., 2003). The relative contributions of these modes to Arctic
sea ice loss remain unclear. In this study, we examine the contribution of the global warming and
PDO modes, whose time coefficients show significant trends, to the Arctic sea ice loss in boreal
summer and autumn during 1979-2017. We will show that the global warming modes in summer
and autumn contribute to 44 and 50%, respectively, of Arctic sea ice loss in these seasons; while
the respective percentages for the PDO mode are 20 and 22%.

**2   Methodology**

The National Snow and Ice Data Center (NSIDC) provides Arctic sea ice concentration data

(http://nsidc.org/data/NSIDC-0051) on a 25 km×25 km grid with a polar stereographic projection
from October 1979 to the present. Although the sea ice data have some defects from surface
flooding (Comiso and Steffen, 2001) and land contamination and weather (Cavalieri et al., 1999),





they can be applicable to the study of changes of Arctic sea ice concentration. The current analyses
use monthly data from boreal summer (June-August) and autumn (September - November).
Atmospheric variables are derived from the European Centre for Medium-Range Weather
Forecasts (ECMWF) ERA-Interim reanalysis (Dee et al., 2011), which has a horizontal resolution
of 79 km (T255) at 60 vertical levels. ERA-Interim reanalysis outperforms other contemporary
global reanalysis datasets, even though it has a warm and moist bias in the planetary boundary
layer (Jakobson et al., 2012). The North Pacific SST patterns are derived from the $2\,°$ latitude $\times\,2\,°$
longitude U.S. National Oceanic and Atmospheric Administration (NOAA) Extended
Reconstructed SST data (http://ftp.cdc.noaa.gov/ noaa.ersst.v5), which is superior in high latitudes
to other SST datasets (Huang et al., 2017).

The empirical orthogonal function (EOF) method is employed to obtain the global warming

and PDO modes considered as the first two modes. The EOF modes include spatial patterns (EOFs)
and corresponding time coefficients or principal components (PCs) characterized with
orthogonality with each other. The global warming signal and the PDO index correspond to the
time series of the first two modes of the SST anomalies in the North Pacific north of 20 °N. The
statistical significance level is tested by the Student's t- test.

**3   Results**
3.1   Arctic sea loss explained by the first two EOF modes

We first present the trends in the North Pacific SST in boreal summer and autumn (Figure 1a

and 1b). Warming trends dominate over the whole study region with significant ones in the
western and central North Pacific. As an important climate mode of the North Pacific, PDO may



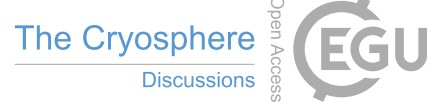

contribute to the warming trend. The PDO indices (http://research.jisao.washington.edu/pdo) show
statistically significant negative trends of -0.0293 ($p < 0.05$) and -0.0261 $yr^{-1}$ ($p < 0.1$) respectively
for boreal summer and autumn (Figure 1c and 1d).

The results of EOF analysis of the North Pacific SST anomalies in boreal summer and autumn

are shown in Figure 2 and 3. The first mode (EOF1) of the summer SST and the second mode
(EOF2) of the autumn SST, explaining 29.6% and 19.6% of total variance, show a nearly uniform
warming pattern in the North Pacific. An increasing trend in the time series for these two EOF
modes (PC1 for summer and PC2 for autumn) represents a global warming mode (Wills et al.,
2018). The warming trends at 0.0623 and 0.0645 per year ($p < 0.05$) for summer and autumn,
respectively,   are not steady with a warming hiatus between 1998 and 2012, flanked by two rapid
warming periods. The domain-averaged warming trends in the North Pacific SST are the same for
summer and autumn, at 0.94 ℃ per century. Trenberth and Shea (2006) considered the global
mean SST as a proxy for external signal. The global mean SST is significantly corrected with
summer PC1 (correlation coefficient 0.79, $p < 0.05$) and autumn PC2 (0.84, $p < 0.05$), suggesting
that these global warming modes in SST is likely to represent an external signal.

The second mode of summer SST and the first mode of autumn SST, accounting for 21.4%

and 27.8% of the total variance for the respective season, represent the positive phase of the PDO
mode, which has negative SST anomalies over the mid-latitudes surrounded by positive SST
anomalies. The time series of these two SST modes, referred as the PDO mode, are highly
correlated with the PDO index with the correlation coefficients of 0.97 between the summer PC2
and PDO and 0.94 between the autumn PC1 and PDO. The PCs of the PDO mode alter from
positive phase with the mean index value of 0.49 before 1998, to negative phase with the mean



value of -0.51 afterwards. The trends in the PCs of the PDO mode are -0.0334 and -0.0349 per
year for summer and autumn (p < 0.05).

Next, we assess the response of Arctic sea ice to the global warming (external) and PDO

(internal) modes, by regressing the Arctic sea ice anomalies onto summer PC1 and autumn PC2
(global warming mode) and to summer PC2 and autumn PC1 (PDO mode) (Figure 4). In both
seasons, the global warming mode is associated with Arctic sea ice loss (Figure 4a and 4d). The
regions with strongest association span the eastern side from Barents Sea to East Siberian,
Chukchi and Beaufort Seas. While the season changes from summer to autumn when Arctic sea
ice is at the minimum value, the region of the largest decrease related to the global warming mode
shifts from the northern Barents Sea to East Siberian and Chukchi Seas. In contrast, the PDO
modes correspond to positive Arctic sea ice anomalies (Figure 4b and 4c). Compared to the global
warming mode, the associations between the PDO mode representing the positive PDO phase and
the Arctic sea ice anomalies are somewhat weaker from Greenland Sea to Beaufort Sea, but
stronger in Baffin Bay, Hudson Bay and the sea near Queen Elizabeth Islands. For both the global
warming mode and the PDO mode, the connection is somewhat stronger in autumn than summer.

Sea ice concentration show a decreasing trend everywhere north of $50^{o}$N except for some

coastal regions of Greenland (Figure 5). Similar to the negative sea ice anomalies related to the
global warming mode in SST that are larger in values in autumn than summer, negative sea ice
trends are also somewhat sharper in autumn than those in summer and the largest negative trends
move from Barents Sea in summer to East Siberian and Chukchi Seas in autumn. The
contributions of the global warming mode and the PDO mode to the total trends in summer and
autumn Arctic sea ice, which is calculated by the product of regression coefficients of sea ice into



the PC (Figure 4) and the trends in the PC (Figures 2 and 3) are shown in Figure 6. Both modes
contribute to Arctic sea ice trends in the two seasons, but the amount of the contribution differs,
with the largest contribution from the autumn global warming mode and the smallest one from the
summer PDO mode. The relative contribution can be also assessed by a contribution ratio
calculated as the ratio of trends explained by the two modes (Figure 6) to the total trends (Figure 5)
and the results at grid points where the trends are significant and the contribution ratio is greater
than 0.001 $yr^{-1}$ are shown in Figure 7. The contribution ratios from the global warming mode are
larger than those from the PDO mode with the exception of Hudson Bay in summer. The
domain-averaged contribution ratios from the global warming mode and the PDO mode are 44.9%
and 20.0%, respectively, in summer and 50.0% and 22.2% in autumn.

3.2    Mechanisms

The relationship between the Arctic sea ice trends and the first two modes of the North Pacific

SST variability merits further consideration in the context of large-scale circulations. Regression
analyses are performed where the 500-hPa geopotential height, mean sea level pressure (MSLP),
850-hPa wind, and surface temperature are regressed into the PCs of the two modes in summer
and autumn and the results are shown in Figures 8-11. In summer, the regression patterns of the
anomalous 500-hPa height and MSLP onto the global warming mode resemble the positive phase
of the NAO and AO (Figure 8a and 9a), which show a nearly barotropic structure. The positive
500-hPa height and MSLP anomalies over the Bering Sea produce an anticyclonic circulation
(Figure 10a), which transports warm air into the Pacific sector of the Arctic, leading to positive
temperature anomalies (Figure 11a) and negative sea ice anomalies there (Figure 4a). The



southerly winds also move the sea ice towards the North Pole, thus resulting in sea ice loss in the
Chukchi Sea. The northerly winds over the northeastern Canada and northern Greenland (Figure
10a) advect warm air to the Kara and Barents Seas, increasing surface air temperature (Figure 11a)
and decreasing sea ice concentration there.

In contrast to summer, the regression pattern in autumn is dominated by positive

500-hPa height anomalies across the Arctic with the exception of northeastern Canada and western
Greenland (Figure 8d). However the anomalous MSLP regression map displays a noticeable
positive phase of the AO index (Figure 9d). The baroclinic structure in autumn differs from the
barotropic feature in summer. The positive MSLP anomalies over the Bering Sea and negative
MSLP anomalies over the Chukchi and East Siberian Seas are favorable for warm air flowing into
the Arctic (Figure 10d), which is related to increasing air temperature (Figure 11d) and decreasing
sea ice over the Pacific sector of the Arctic (Figure 4d). The negative MSLP anomalies over
Greenland and positive MSLP anomalies over Northern Europe induce southwesterly winds over
North Atlantic Ocean extending to most of the Arctic resulting in more significant warming and
Arctic sea ice loss in autumn than in summer. Although the anomalous North Pacific SST patterns
related to the global warming mode are similar in summer and autumn, the corresponding
atmospheric circulations patterns are different, and produce noticeable differences in the pattern of
surface air temperature increases and sea ice loss in the Arctic.

In boreal summer, the positive phase of the PDO mode is related to a Rossby wavetrain

extending from the North Pacific and North America to the Arctic Ocean and Europe (Figure 8b).
Throughout the Arctic, negative anomalies in 500-hPa height and MSLP dominate, corresponding
to slightly positive phase of the AO index (Figure 9b). The anomalous southerly winds induced by



the negative MSLP over Greenland produce negligible warming in the northern North Atlantic and
central Arctic (Figure 10b). On the contrary, northerly winds from the North Pole generate
significant cooling in terrestrial Arctic and northeastern Canadian archipelago (Figure 11b), where
sea ice concentration increases significantly (Figure 4b). Meanwhile the northerly winds drive the
sea ice into the surrounding seas, leading to the increase in sea ice concentration there.
In autumn, the wavetrain occurs over the North Pacific, North America, and North Atlantic
(Figure 8c). The positive MSLP anomalies produce increasing (decreasing) air temperature and
decreasing (increasing) sea ice over Greenland and the Greenland Sea (Barents Sea), related to
anomalous southerly (northerly) winds (Figure 9c, 10c, 11c and 4c). Over the Laptev and East
Siberian Seas, anomalous northerly winds also generate significant cooling and sea ice increase.
The anomalous high moves from Bering Strait to the Gulf of Alaska, which limits the warming
into the Arctic Ocean. Thus the Pacific sector of the Arctic shows a cooling tendency and
increasing sea ice concentration. Similar to the global warming mode, the PDO mode also shows a
seasonal feature in its effect on atmospheric circulation and sea ice with more significant influence
in autumn than summer. The response of atmospheric circulation to the PDO mode shows a more
barotropic structure than the response to the global warming mode.

**4   Discussion and Conclusions**
Following the suggestion that the North Pacific SST anomalies play an important role in the
melt season Arctic sea ice loss (Yu and Zhong, 2018), the current study further assesses the
relative contribution of the two leading EOF modes in SST variability, representing the global
warming (external) and PDO (internal) modes, to the trends in Arctic sea ice in boreal summer and



autumn for the recent four decades (1979-2017). As the first two modes of the North Pacific SST
variability, the time coefficients of the global warming (summer PC1 and autumn PC2) and the
PDO (summer PC2 and autumn PC1) modes exhibit a significant increasing and decreasing trend,
respectively. In summer, the PDO and global warming modes contribute to 20.0% and 44.9% of
Arctic sea ice loss, respectively; while in autumn the percentages are 22.2% and 50.1%. Both
modes also exert more significant effects on large-scale atmospheric circulations in autumn than in
summer. The response of corresponding atmospheric circulations to the two modes also differs in
summer and autumn, especially over northern North Atlantic. In contrast to summer, the autumn
anomalous atmospheric circulations related to the global warming mode are more baroclinic. For
the PDO mode, the wavetrain propagates more eastwards in summer than in autumn. The
anomalous surface wind fields related to the two modes perturb the dynamic and thermodynamic
environments in ways that are consistent with the observed patterns of the Arctic sea ice change.

Previous studies investigating the contributions of external and internal forcings to Arctic sea

ice loss have been based heavily on numerical modeling. Model results, however, have shown
large departures from observations in the Arctic due to the lack of understanding in sea ice
dynamics and thermodynamics and their interactions with the atmosphere and other uncertainties
in physical parameterizations and numerical algorithms (Winton, 2011; Stroeve et al., 2012;
Mahlstein and Knutti, 2012). The results here are based on reanalysis products which are
considered more reliable than model outputs because of the assimilation of in-situ observations
and remote sensing satellite data. Previous studies have suggested that internal forcing may
explain somewhere between 20% to 50% of Arctic sea ice loss (Stroeve et al., 2007; Kay et al.,
2011; Day et al., 2012; Ding et al., 2019). Our results show that internal forcing represented by the

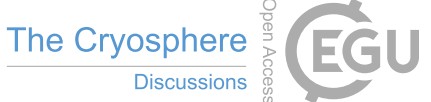

PDO mode contributes to slightly more than 20% of the Arctic sea ice loss in summer and autumn
and thus total contribution from internal factors must exceed 20%.
In addition to PDO, the AMO mode is also found to be important to the Arctic sea ice loss
through its effect on oceanic and atmospheric heat transport (Yu et al., 2017; Zhang, 2015). Day et
al. (2012) attributed 5-30% of Arctic sea loss to the AMO mode. It must be cautioned that the parts
of the global warming mode should be removed when estimating the contribution of the AMO
mode to the Arctic sea ice loss (Ting et al., 2009). Besides SST in the North Pacific and Atlantic,
other important factors for Arctic sea ice loss in summer and autumn include the effects of
atmospheric internal variability on heat and moisture transports from mid-latitudes to the Arctic
(Kapsch et al., 2013; Naakka et al., 2019).
In this study, the contribution of the global warming mode to the Arctic sea ice depletion is
explained in the context of atmospheric circulation anomalies. The effect of the global warming
mode also work directly through some local feedback processes (Vihma et al., 2014), including
ice-albedo feedback (Flanner et al., 2011), water vapor and cloud-radiative feedback (Sedlar et al.,
2011), and processes related to lower atmosphere stability such as surface inversion (Bintanja et al.
2011; Pithan and Mauritsen, 2014). The external forcing also may interact with the
above-mentioned internal forcing (Ding et al., 2019). The global warming mode considered here
combines all anthropogenic factors, including greenhouse gas, aerosols, and ozone. The data and
analysis tools used in this study are unable to separate their individual contributions.







*Author contributions*. LY designed the study and analyzed the data. All authors discussed the
results and contributed to the writing and editing of the manuscript.
*Competing interests*. All authors declare that they have no conflict of interest.
*Acknowledgements*. We thank the agencies for providing the datasets used in this study. This work
was supported by the National Key R&D Program of China (No. 2017YFE0111700) and the
Academy of Finland (contract 317999).


















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




Figure captions
Figure 1. The trends in the North Pacific SST in summer (a) and autumn (b) ($^{o}$C yr$^{-1}$) and time
series of the PDO indices in summer (c) and autumn (d) for the period 1979-2017. Dotted regions
in Figure (a) and (b) indicate above 95% confidence level. Dashed lines in Figure (c) and (d)
denote the trends in the PDO indices.
Figure 2. Spatial patterns (EOF1 and EOF2) and time series (PC1 and PC2) of the leading two
EOF modes of summer North Pacific SST over the region (120°E-100°W, 20°N-65°N) during
1979-2017. The number in the left panels indicates the percentage of variance explained by the
two modes. The black dashed lines in the right panels denote the trends for the period 1979-2017.
Figure 3. The same as Figure 2, but for autumn.
Figure 4. Regression maps of summer (a), (b) and autumn (c), (d) sea ice concentration anomalies
into the time series of the first (a), (c) and second (b), (d) mode of summer (a), (b) and autumn (c),
(d) SST anomalies in the North Pacific. Dotted regions denote above 95% confidence level.
Figure 5. Trends in sea ice concentration (yr$^{-1}$) for summer (a) and autumn (b). Dotted regions
denote above 95% confidence level.
Figure 6. Trends in sea ice concentration (yr$^{-1}$) explained by the first (a), (c) and second (b), (d)
modes of summer (a), (b) and autumn (c), (d) North Pacific SST anomalies.
Figure 7. The ratio of trends explained by the first (a), (c) and second (b), (d) modes of summer (a),
(b) and autumn (c), (d) North Pacific SST anomalies. Only grid points where the trends are
significant and more than 0.001 yr$^{-1}$ are shown.
Figure 8. Regression maps of 500-hPa geopotential height (gpm) onto the time series of the first
(a), (c) and second (b), (d) mode of summer (a), (b) and autumn (c), (d) North Pacific SST





anomalies. Dotted regions indicate above 95% confidence level.
Figure 9. The same as Figure 8, but for mean sea level pressure (MSLP) (Pascal).
Figure 10. The same as Figure 8, but for 850-hPa wind field.
Figure 11. The same as Figure 8, but for surface air temperature ($^{\circ}$C).






















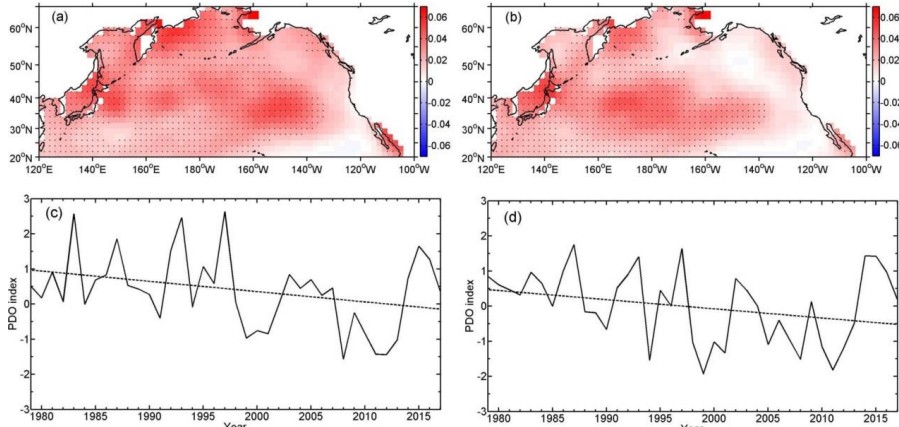

Figure 1. The trends in the North Pacific SST in summer (a) and autumn (b) (oC yr-1) and time series of the PDO indices in summer (c) and autumn (d) for the period 1979-2017. Dotted regions in Figure (a) and (b) indicate above 95% confidence level. Dashed lines in Figure (c) and (d) denote the trends in the PDO indices.






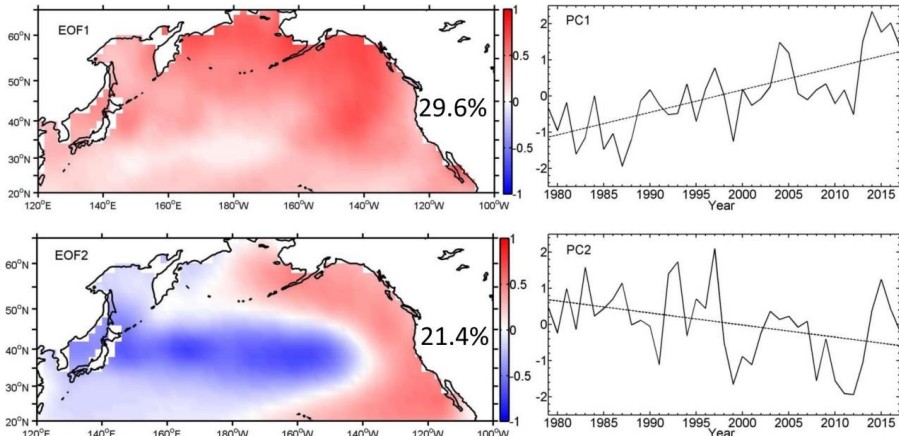

Figure 2. Spatial patterns (EOF1 and EOF2) and time series (PC1 and PC2) of the leading two
EOF modes of summer North Pacific SST over the region (120°E-100°W, 20°N-65°N) during
1979-2017. The number in the left panels indicates the percentage of variance explained by the
two modes. The black dashed lines in the right panels denote the trends for the period 1979-2017.





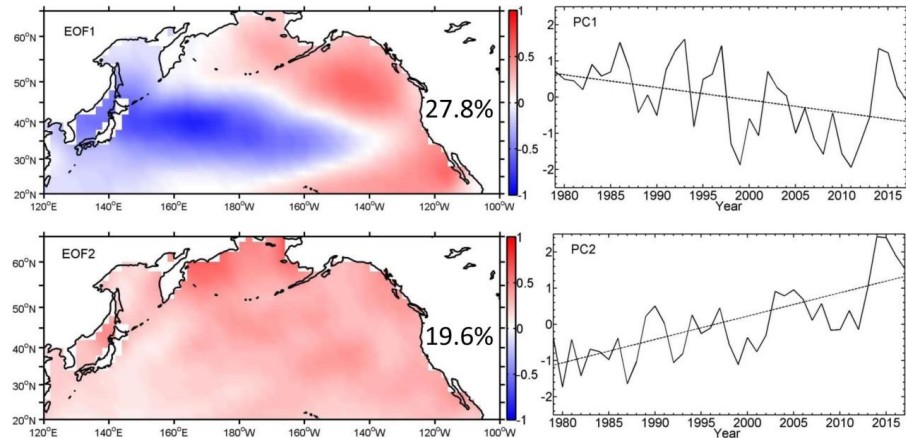

Figure 3. The same as Figure 2, but for autumn.





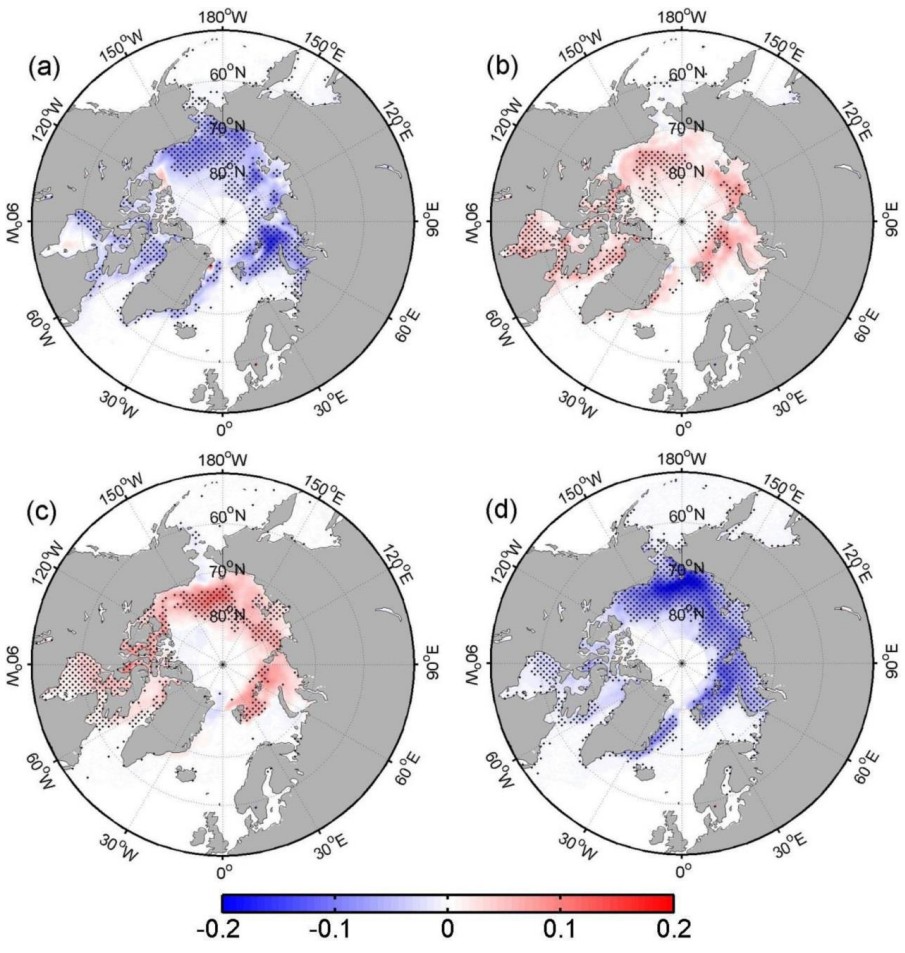

Figure 4. Regression maps of summer (a), (b) and autumn (c), (d) sea ice concentration anomalies into the time series of the first (a), (c) and second (b), (d) mode of summer (a), (b) and autumn (c), (d) SST anomalies in the North Pacific. Dotted regions denote above 95% confidence level.





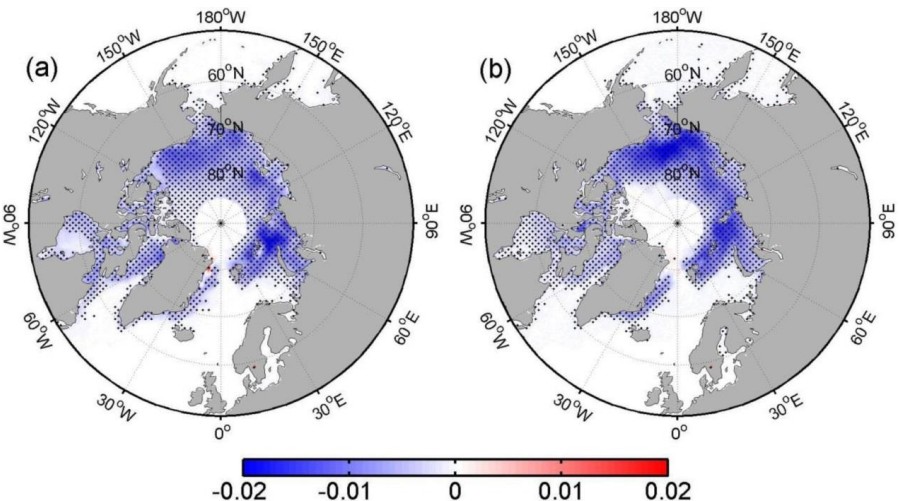


Figure 5. Trends in sea ice concentration (yr$^{-1}$) for summer (a) and autumn (b). Dotted regions
denote above 95% confidence level.































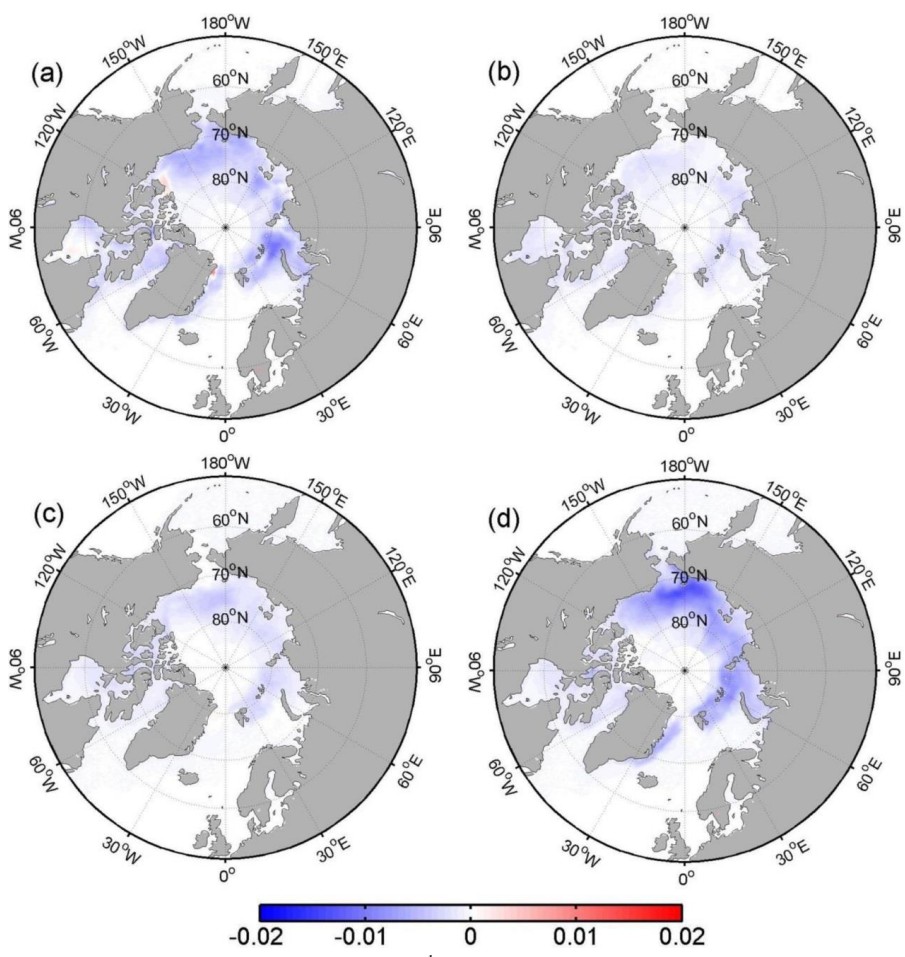

Figure 6. Trends in sea ice concentration (yr$^{-1}$) explained by the first (a), (c) and second (b), (d)
modes of summer (a), (b) and autumn (c), (d) North Pacific SST anomalies.

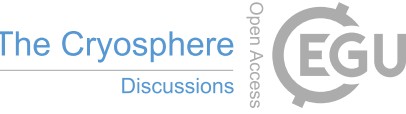



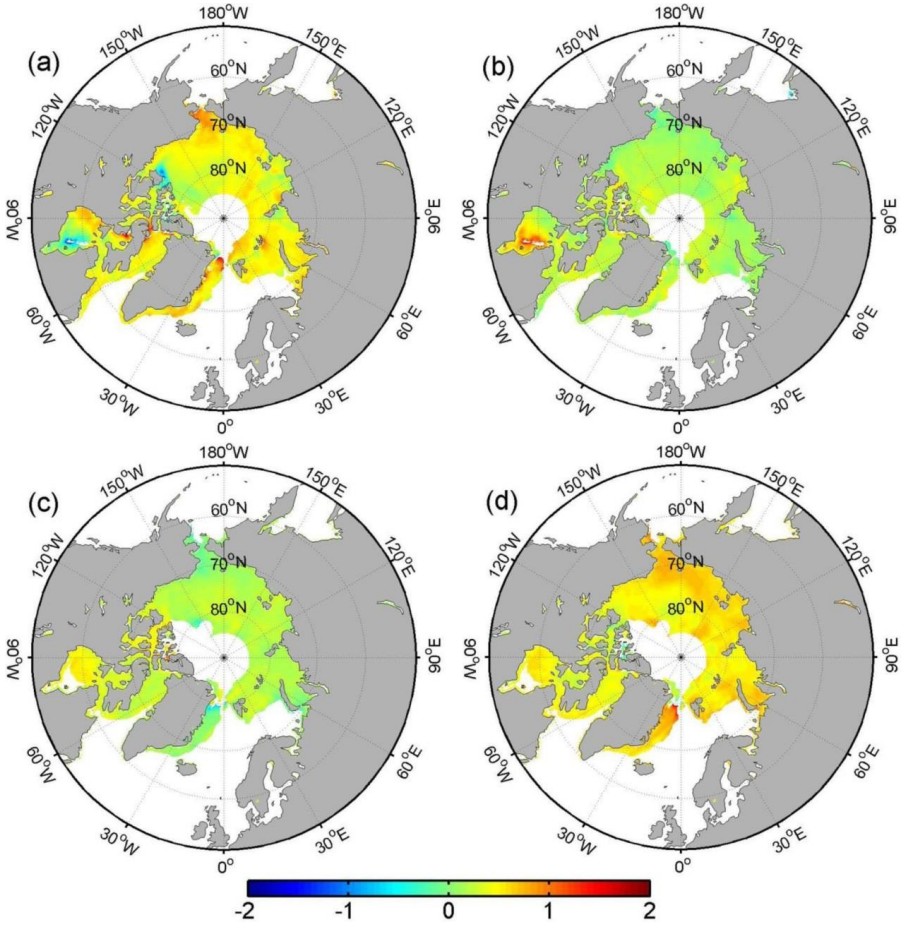


Figure 7. The ratio of trends explained by the first (a), (c) and second (b), (d) modes of summer (a), (b) and autumn (c), (d) North Pacific SST anomalies. Only grid points where the trends are significant and more than 0.001 yr$^{-1}$ are shown.




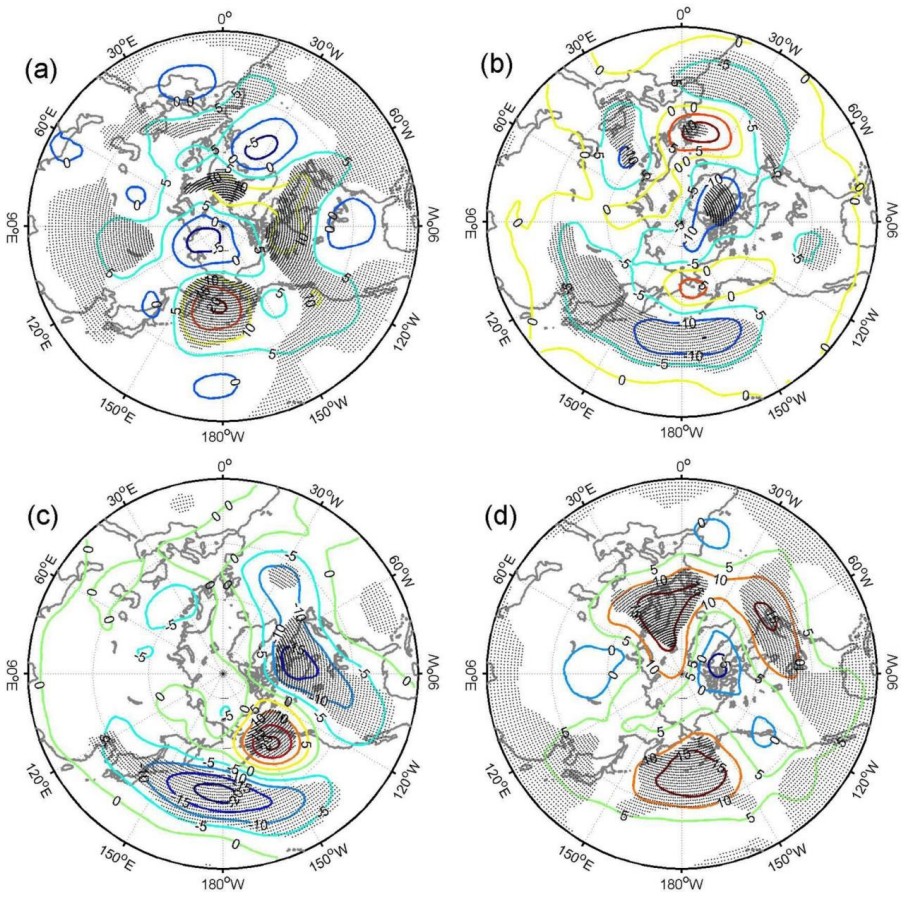

Figure 8. Regression maps of 500-hPa geopotential height (gpm) into the time series of the first (a),
(c) and second (b), (d) mode of summer (a), (b) and autumn (c), (d) North Pacific SST anomalies.
Dotted regions indicate above 95% confidence level.




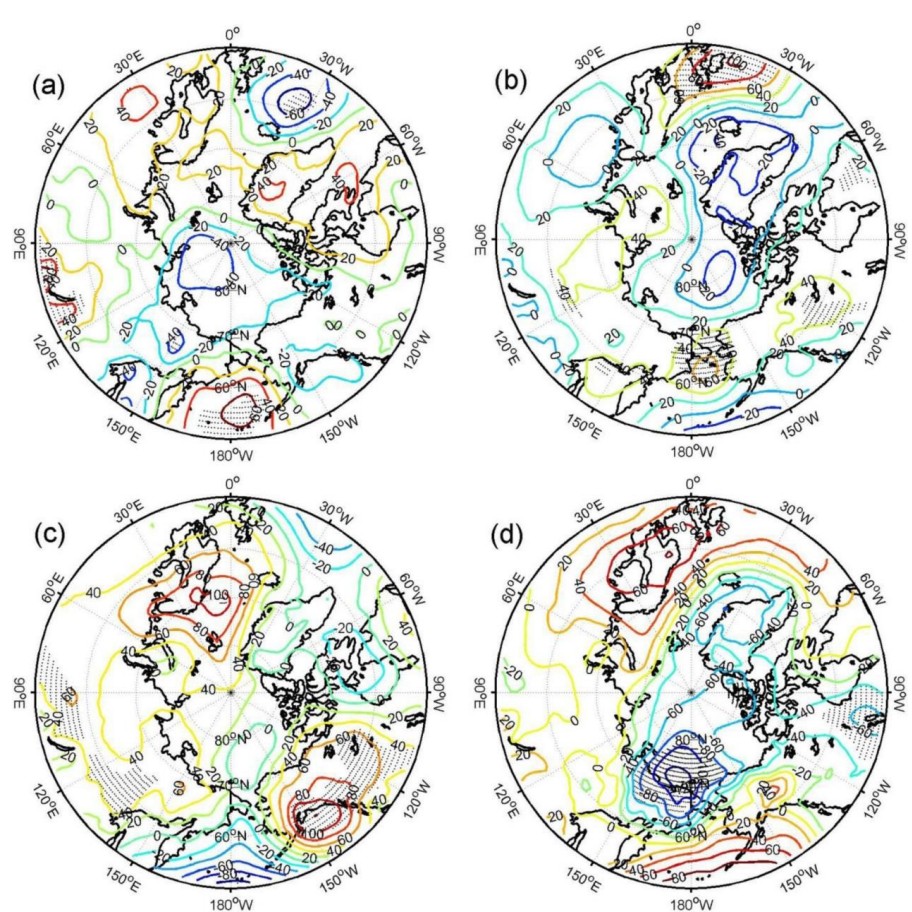


Figure 9. The same as Figure 8, but for mean sea level pressure (MSLP) (Pascal).




















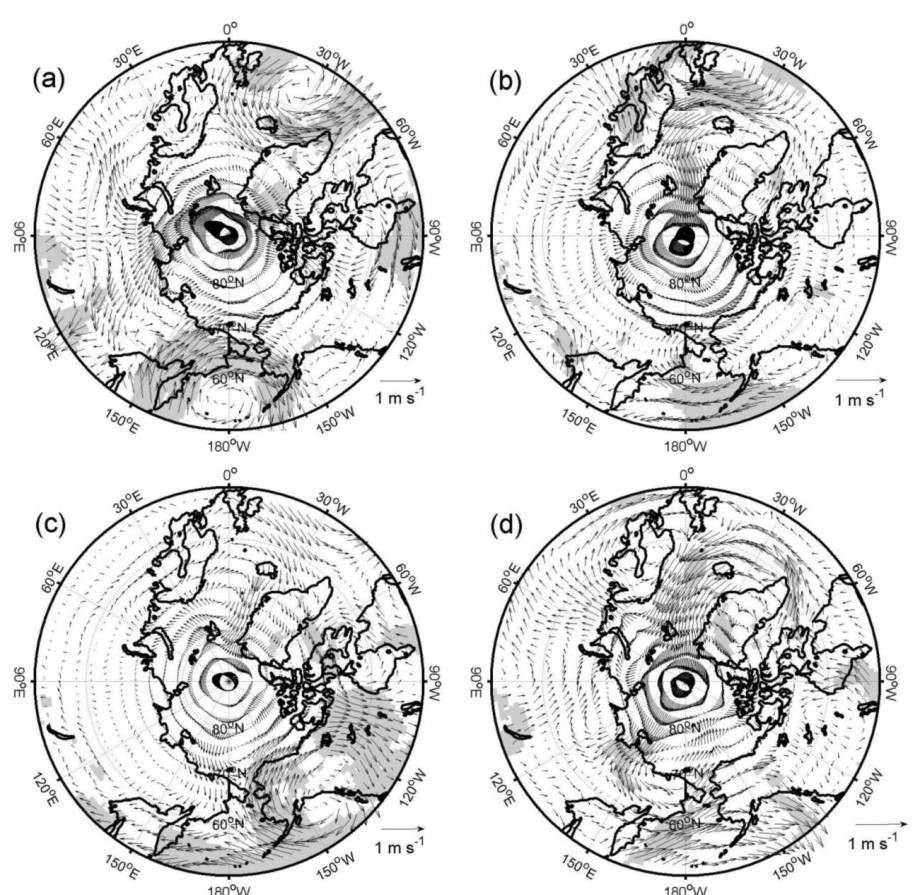


Figure 10. The same as Figure 8, but for 850-hPa wind field.



















Figure 11. The same as Figure 8, but for surface air temperature ($^{o}$C).
