# Peer review of "The contributions of the leading modes of the North Pacific sea surface temperature variability to the Arctic sea ice depletion in recent decades"

_The Cryosphere, 2019_

## Referee Comment (RC1) · Anonymous Referee #1 · 18 Jun 2019

Peer review of "The contributions of the leading modes of the North Pacific sea surface temperature variability to the Arctic sea ice depletion in recent decades" by L. Yu et al.

This work considers the effect of internal climate variability from the Pacific Decadal Oscillation (PDO) and background global warming on Arctic sea-ice retreat. The two leading empirical orthogonal functions (EOFs) of the sea-surface temperatures in the North Pacific are estimated and argued to represent PDO and global warming respectively. The Arctic sea-ice extent is regressed onto these modes and the portion of variability and trends of the sea-ice extent explained by these modes are estimated. In order to study physical processes in the coupling between the North Pacific temperature trends and Arctic sea ice, also fields of geopotential height, surface pressure, wind and temperatures are regressed onto the two leading North Pacific SST EOFs.

The EOF and regression approaches are simple, and the work clearly shows that Arctic sea-ice decline is coupled to both global warming and and long-term change of the PDO phase representing a given mode of internal variability in the climate system. However the attempt to explore the physical processes in the coupling between the North Pacific SST EOFs and the Arctic sea ice is little convincing. Below I have some suggestions for improvements. I recommend major revisions.

Specific comments

I am critical to the arguments presented in the section Mechanisms:

1) Line 173-175: Based on Figs. 8a and 9a it is claimed that "the global warming mode resemble the global warming mode of NAO and AO. I couldn't see this from the figures, and the lack of connection is also consistent with no general trend in the AO and NAO indices over the last decades.

2) Line 175-178: It is here claimed that an anticyclone in the Bering sea in summer brings warm air into the Arctic causing ice melt. The reader is lead to Fig. 11a for the warming pattern, but I couldn't find it.

Most of the section continues with discussing patterns in the regression plots that are difficult or not possible to see. The choice of variables may not be appropriate for studying the physical processes relevant for the coupling between the EOF modes and the Arctic sea ice. Here are two suggestions:

i) Temperature anomalies are not appropriate when investigating effect of advection over sea ice in summer, since convergence of energy associated with the advection often goes directly into sea-ice melt rather than warming. More appropriate variables would be the surface energy budget, and water vapour and clouds, where the later two are coupled to the changes of the greenhouse effect over sea ice. Also the greenhouse

effect in itself can be estimated, as the difference between outgoing longwave radiation at the top of the atmosphere and the surface.

ii) Apply lagged regression with a daily resolution in order to study cause and effect between variables.

Note that PDO variability may not cause sea-ice variability even though the two appear related as shown by the regressions. Another process may cause both warming of the North Pacific and melting of the Arctic sea ice at the same time.

Small suggestions and typos:

Line 25: "The" in front of "Arctic", move "sea ice" to after "decrease in".

Line 26-29: This sentence is difficult to read and should be reformulated.

Line 48: "decline" -> "declining".

Line 69: Comma after "forcings".

Line 79-82: This sentence also needs a reformation.

Line 125: "corrected" -> "correlated".

Line 135: What are the units of the numbers mentioned in this line?

Line 150: "The" in front of "sea ice", "shows" with "s"

Line 153: "sharper" -> "stronger/larger"

Line 156: "into" -> "onto", and many other places.

Line 160: "also" before "be".

Fig 7.: The colour scaling is not so well chosen, I suggest to let the scale goes from -1 to 1.

Fig. 8-11: What is the reason why these figures are shown upside down relative to Fig.

7?

---

## Editor Comment (EC1) · Jennifer Hutchings (Editor) · 8 Jul 2019

Review of
**"The contributions of the leading modes of the North Pacific sea surface temperature variability to the Arctic sea ice depletion in recent decades"**
by Yu et al.

This study investigates the Pacific SST modes and possible relationship to Arctic summer ice cover. The first mode is global warming mode, while the 2nd the PDO mode. Then the EOF coefficients are regressed to 500hPa height, SLP, wind, and SAT fields to obtain the regression maps. The conclusions and discussion are based on these regression results. Some of the results are well known and expected, basically descriptive or qualitative. Dynamical and thermodynamical explanations are laced, and the conclusions are not convincing.

**Majors:**
1) September ice data should be considered summer in the Arctic, because of the minimum ice cover, see many others' definition: winter: month 1, 2,3; spring: 4,5,6; summer: 7,8,9; fall: 10,11,12
2) First EOF SST mode is very similar to the 2nd mode in space and time (Fig. 2), except weaker in space. If you calculate the coef between the two, it should be correlated (please do so). Then, the question is these two modes may not be separated, 30% vs 22% or at least they are correlated. Please test they can be separated using North et l (1982)
3) If the same EOF is applied to the North Atlantic, is the 1st mode the same as this Pacific global warming mode because it is defined as the GLOBAL Warming? Or the AMO is major forcing from the Atlantic side with the same warming trend?
4) When the global mode is repressed to SLP, an AO/NAO mode appears (Fig. 8 and related discussion), this indicates the so-called the global mode is something related to AO/NAO, which has a long-term trend. Therefore, only using PDO and global warming modes can be very misleading, because there are ENSO mode, and AO/NAO and AMO modes from the Atlantic side. Therefore, it is not that simple; and the conclusions are not convincing
5) The work is basically qualitative. To describe the mechanisms, the oceanic heat transport from both the Pacific and Atlantic, and atmospheric heat transport into/out of Arctic may be calculated, and correlated to global and PDO modes
6) Fig. 4. The sea ice concentration varies from 0 to 10 or 0 to 100%, which is not a random variables, strictly speaking. This is why the changes can be seen only on the first year ice area, rather than the central Arctic (white area). Actually, the sea ice there also experiences significant decline.
7) Fig. 9. The AO mode here is not a typical AO mode, it is more like a combined AO and other modes not investigated here.

Based on my professional training and knowledge, I would not recommend it to be published in the Cryosphere, because of above concerns and confusions.

**Minors:**
English needs polishing throughout

Fig. 10. The anomalous wind fields are difficult to digest.
Fig. 3. Are the two modes separated?

---

## Author Comment (AC1) · 8 Jul 2019

Specific comments I am critical to the arguments presented in the section Mechanisms: 1) Line 173-175: Based on Figs. 8a and 9a it is claimed that "the global warming mode resemble the global warming mode of NAO and AO. I couldn't see this from the figures, and the lack of connection is also consistent with no general trend in the AO and NAO indices over the last decades. 2) Line 175-178: It is here claimed that an anticyclone in the Bering sea in summer brings warm air into the Arctic causing ice melt. The reader is lead to Fig. 11a for the warming pattern, but I couldn't find it. Most of the section continues with discussing patterns in the regression plots that are difficult or

not possible to see. The choice of variables may not be appropriate for studying the physical processes relevant for the coupling between the EOF modes and the Arctic sea ice. Here are two suggestions: i) Temperature anomalies are not appropriate when investigating effect of advection over sea ice in summer, since convergence of energy associated with the advection often goes directly into sea-ice melt rather than warming. More appropriate variables would be the surface energy budget, and water vapour and clouds, where the later two are coupled to the changes of the greenhouse effect over sea ice. Also the greenhouse effect in itself can be estimated, as the difference between outgoing longwave radiation at the top of the atmosphere and the surface. ii) Apply lagged regression with a daily resolution in order to study cause and effect between variables.

Response: Following the suggestion, we have added analyses of the variables that are appropriate for studying physical processes relevant for coupling between the EOF modes and the Arctic sea ice. These variables include surface net solar radiation, downward longwave radiation, and total column water vapor and the results are discussed in a new section (3.2.2.) with three new figures (Figures 12-14). We have also estimated the greenhouse effect based on the difference between outgoing longwave radiation at the top of the atmosphere and the surface, which are significantly correlated with the time coefficients of the warming modes for summer and autumn with the correlation coefficients of -0.35 and -0.60, respectively. The result are shown below (Figures 1s) but not in the manuscript.

Note that PDO variability may not cause sea-ice variability even though the two appear related as shown by the regressions. Another process may cause both warming of the North Pacific and melting of the Arctic sea ice at the same time.

Response: To acknowledge this, we have added a following paragraph to the end of the paper Before ending, a word of caution about the PDO mode is in order here. A statistically significant connection between the PDO mode and the Arctic sea ice variability found in this analysis does not necessarily lead to the conclusion that PDO

causes the Arctic sea ice loss unless confirmed by modeling studies focusing on the underlying physical mechanisms. It is possible that another process may cause both the warming of the North Pacific and the melting of the Arctic sea ice at the same time.

Small suggestions and typos: Line 25: "The" in front of "Arctic", move "sea ice" to after "decrease in". Changed

Line 26-29: This sentence is difficult to read and should be reformulated. Reformulated

Line 48: "decline" -> "declining". Changed

Line 69: Comma after "forcings". Added

Line 79-82: This sentence also needs a reformation. Reformed

Line 125: "corrected" -> "correlated". Changed

Line 135: What are the units of the numbers mentioned in this line? The time series of the first two EOF modes is standardized. There is no unit.

Line 150: "The" in front of "sea ice", "shows" with "s" Added

Line 153: "sharper" -> "stronger/larger" Changed

Line 156: "into" -> "onto", and many other places. Changed

Line 160: "also" before "be". Changed

Fig 7.: The colour scaling is not so well chosen, I suggest to let the scale goes from -1 to 1. Changed

Fig. 8-11: What is the reason why these figures are shown upside down relative to Fig. 7? Figures 4-7 has been modified to be consistent with Figures 8-11 in the geographical location.

Please also note the supplement to this comment:
https://www.the-cryosphere-discuss.net/tc-2019-38/tc-2019-38-AC1-supplement.zip

[Figure]

(a) (b) (c) (d)

Fig. 1.

**Fig. 2.**

(a)

(b)

(c)

(d)

**Fig. 3.**

The difference between domain-averaged outgoing longwave radiation north of $70°$N at the top of the atmosphere and the surface for summer (a) and autumn (b)

[Figure]

**Fig. 4.**

---

## Author Comment (AC2) · 15 Jul 2019

We hope that our answers and new analyses have addressed the reviewer's concerns

Please also note the supplement to this comment:
https://www.the-cryosphere-discuss.net/tc-2019-38/tc-2019-38-AC2-supplement.pdf

[Figure]

**Fig. 1.**

[Figure]

**Fig. 2.**

[Figure]

**Fig. 3.**

The difference between domain-averaged outgoing longwave radiation north of $70°$N at the top of the atmosphere and the surface for summer (a) and autumn (b)

[Figure]

**Fig. 4.**

[Figure]

**Fig. 5.**

[Figure]

**Fig. 6.**

**Supplement:**

Reviewer 1

Specific comments

I am critical to the arguments presented in the section Mechanisms:

1) Line 173-175: Based on Figs. 8a and 9a it is claimed that "the global warming mode resemble the global warming mode of NAO and AO. I couldn't see this from the figures, and the lack of connection is also consistent with no general trend in the AO and NAO indices over the last decades.

2) Line 175-178: It is here claimed that an anticyclone in the Bering sea in summer brings warm air into the Arctic causing ice melt. The reader is lead to Fig. 11a for the warming pattern, but I couldn't find it.

Most of the section continues with discussing patterns in the regression plots that are difficult or not possible to see. The choice of variables may not be appropriate for studying the physical processes relevant for the coupling between the EOF modes and the Arctic sea ice. Here are two suggestions:

    i)        Temperature anomalies are not appropriate when investigating effect of advection over sea ice in summer, since convergence of energy associated with the advection often goes directly into sea-ice melt rather than warming. More appropriate variables would be the surface energy budget, and water vapour and clouds, where the later two are coupled to the changes of the greenhouse effect over sea ice. Also the greenhouse effect in itself can be estimated, as the difference between outgoing longwave radiation at the top of the atmosphere and the surface.

    ii)      Apply lagged regression with a daily resolution in order to study cause and effect between variables.

Following the suggestion, we have added analyses of the variables that are appropriate for studying physical processes relevant for coupling between the EOF modes and the Arctic sea ice. These variables include surface net solar radiation, downward longwave radiation, and total column water vapor and the results are discussed in a new section (3.2.2.) with three new figures (Figures 12-14). We have also estimated the greenhouse effect based on the difference between outgoing longwave radiation at the top of the atmosphere and the surface, which are significantly correlated with the time coefficients of the warming modes for summer and autumn with the correlation coefficients of -0.35 and -0.60, respectively. The result are shown below (Figures 1s) but not in the manuscript.

[Figure]

Figure 1s The difference between outgoing longwave radiation at the top of the atmosphere and the surface for summer (a) and autumn (b).

3.2.2 Radiation and water vapor feedback

Besides changes in large-scale atmospheric circulation, the two leading modes in the North Pacific SST anomalies, especially the warming mode, can also influence Arctic sea ice variability through other mechanisms such as albedo feedback and water vapor feedback. To understand the role these mechanisms play in the variability of Arctic sea ice in boreal summer and autumn, regression analysis is also performed where surface net solar radiation, surface downward longwave radiation and total column water vapor are regressed onto the time series of the two modes for boreal summer and autumn. Across the Arctic Ocean, the sea ice anomalies are inversely related to the net solar radiation anomalies (Figure 12) and the magnitudes of the anomalous net solar radiation is larger in summer than autumn, suggesting that the albedo feedback mechanism, which is related more closely to the warming mode than the PDO mode, is likely to play a more important role in Arctic sea ice loss in summer than autumn (Figure 12).

Similar to net solar radiation, positive (negative) downward longwave radiation anomalies are associated with decreased (increased) Arctic sea ice (Figure 13). The anomalous downward longwave radiation are over the Atlantic sector of the Arctic Ocean in summer, extending to cover the entire Arctic Ocean in autumn, suggesting larger influence of water vapor feedback on Arctic sea ice melt in autumn than summer. Because of the direct influence of water vapor on longwave radiation, it is no surprising to find that the spatial regression pattern of the anomalous total column water vapor (Figure 14) is similar to that of the anomalous downward longwave radiation. The increases in water vapor can be linked partially to the temperature increase due to increased greenhouse gas emissions and partially to water vapor transport from lower latitudes. For example, the anomalous southerly winds over the northern Atlantic Ocean and the Bering Strait can transport water vapor from lower latitude into the Arctic Ocean (Figure 10d).

[Figure]

Figure 12. The same as Figure 8, but for surface net solar radiation (W m$^{-2}$).

[Figure]

Figure 13. The same as Figure 8, but for surface downward longwave radiation (W m$^{-2}$).

[Figure]

Figure 14. The same as Figure 8, but for total column water vapor (kg m$^{-2}$).

Note that PDO variability may not cause sea-ice variability even though the two appear related as shown by the regressions. Another process may cause both warming of the North Pacific and melting of the Arctic sea ice at the same time.

To acknowledge this, we have added a following paragraph near the end of the paper

Before ending, a word of caution about the PDO mode is in order here. A statistically significant connection between the PDO mode and the Arctic sea ice variability found in this analysis does not necessarily lead to the conclusion that PDO causes the Arctic sea ice loss unless confirmed by modeling studies focusing on the underlying physical mechanisms. It is possible that another process may cause both the warming of the North Pacific and the melting of the Arctic sea ice at the same time.

Small suggestions and typos:

Line 25: "The" in front of "Arctic", move "sea ice" to after "decrease in".

Changed

Line 26-29: This sentence is difficult to read and should be reformulated.

Reformulated

Line 48: "decline" -> "declining".

Changed

Line 69: Comma after "forcings".

Added

Line 79-82: This sentence also needs a reformation.

Reformed

Line 125: "corrected" -> "correlated".

Changed

Line 135: What are the units of the numbers mentioned in this line?

The time series of the first two EOF modes is standardized. There is no unit.

Line 150: "The" in front of "sea ice", "shows" with "s"

Added

Line 153: "sharper" -> "stronger/larger"

Changed

Line 156: "into" -> "onto", and many other places.

Changed

Line 160: "also" before "be".

Changed

Fig 7.: The colour scaling is not so well chosen, I suggest to let the scale goes from -1 to 1.

Changed

Fig. 8-11: What is the reason why these figures are shown upside down relative to Fig. 7?

Figures 4-7 has been modified to be consistent with Figures 8-11 in the geographical location.

Reviewer 2

This study investigates the Pacific SST modes and possible relationship to Arctic summer ice cover. The first mode is global warming mode, while the 2nd the PDO mode. Then the EOF coefficients are regressed to 500hPa height, SLP, wind, and SAT fields to obtain the regression maps. The conclusions and discussion are based on these regression results. Some of the results are well known and expected, basically descriptive or qualitative. Dynamical and thermodynamical explanations are laced, and the conclusions are not convincing.

Majors:

1) September ice data should be considered summer in the Arctic, because of the minimum ice cover, see many others' definition: winter: month 1, 2,3; spring: 4,5,6; summer: 7,8,9; fall: 10,11,12

The reviewer is absolutely correct that September sea ice should be considered summer. The summer and autumn definition here are based on North Pacific sea surface temperature. This has been clarified in the manuscript "This analysis uses monthly data from boreal summer and autumn defined as June through August and September through November, respectively, based on North Pacific sea surface temperature."

First EOF SST mode is very similar to the 2nd mode in space and time (Fig. 2), except weaker in space. If you calculate the coef between the two, it should be correlated (please do so). Then, the question is these two modes may not be separated, 30% vs 22% or at least they are correlated. Please test they can be separated using North et l (1982)

Although first EOF seems similar to second EOF, the correlation of their time coefficients is zero due to the orthogonality of EOF analysis.

Using the method of North et al. (1982), $29.6-21.4=8.2>6.7=29.6\times\left(\dfrac{2}{39}\right)^{0.5}$ the two modes are separated.

2) If the same EOF is applied to the North Atlantic, is the 1st mode the same as this Pacific global warming mode because it is defined as the GLOBAL Warming? Or the AMO is major forcing from the Atlantic side with the same warming trend?

The first mode of North Atlantic SST is the AMO mode (Figure 2s). The correlation coefficient between the PC1 and the summer AMO index (http://www.esrl.noaa.gov/psd/data/timeseries/AMO/) is 0.83. The correlation coefficient between the PC1 and the autumn AMO index is 0.93. The time coefficients of the first mode of North Atlantic SST show an abrupt change in the late 1990s, not a gradual increasing trend like the global warming.

[Figure]

Figure 2s Spatial patterns (EOF1) and time series (PC1) of the leading two EOF modes of summer and autumn North Atlantic SST over the region (80°W-0°W, 20°N-70°N) during 1979-2017.

3) When the global mode is repressed to SLP, an AO/NAO mode appears (Fig. 8 and related discussion), this indicates the so-called the global mode is something related to AO/NAO, which has a long-term trend. Therefore, only using PDO and global warming modes can be very misleading, because there are ENSO mode, and AO/NAO and AMO modes from the Atlantic side. Therefore, it is not that simple; and the conclusions are not convincing

The reviewer is correct that ENSO, AO/NAO and AMO modes may influence Arctic sea ice change. We calculated the trends in AO/NAO and ENSO for summer and autumn and noted that only the trend in summer NAO is significantly negative and is related to the summertime global mode. AMO plays an important role in Arctic sea ice loss. We discussed its role in the Discussion section. In this study, we only highlighted the effects of global warming and PDO modes (Pacific SST change) on Arctic sea ice loss.

4) The work is basically qualitative. To describe the mechanisms, the oceanic heat transport from both the Pacific and Atlantic, and atmospheric heat transport into/out of Arctic may be calculated, and correlated to global and PDO modes

Previous studies, such as Zhang (2015), have examined the contributions of Atlantic and Pacific Ocean heat transport into the Arctic to Arctic sea ice loss. Thus, here we only consider the effect of atmospheric heat transport on Arctic sea ice loss. We calculate the northward total energy flux at 70°N and regress it onto the time series of the global and PDO modes for summer and autumn (Figure 15). A new figure (Figure 15) is now added to the manuscript along with a new section (3.2.3) to discuss the effect of atmospheric heat transport on Arctic sea ice loss.

[Figure]

Figure 15 Regression of the northward total energy flux at 70 N onto the PC1 (a), (c) and PC2 (b), (d) for summer (a), (b) and autumn (c), (d). Red lines indicate the above 95% confidence level.

3.2.3 Atmospheric heat transport

Atmospheric heat transport from lower latitudes into the Arctic Ocean can also contribute to Arctic sea ice change. This effect is examined by regressing the northward total energy flux at 70 N onto the time series of the warming and PDO modes for summer and autumn (Figure 15). Positive heat transport anomalies correspond to anomalous 850-hPa southerly winds and the opposite occurs for anomalous northerly winds (Figure 10). For summer warming mode, the largest heat transport into the Arctic occurs over the eastern Siberia, followed by those over northern Canada and northern Atlantic Ocean; In contrast, heat is transported out of the Arctic over the Kara Sea, northeastern Canada and Alaska. For summer PDO mode, the heat transport is weaker over the eastern Siberia and somewhat stronger over northern Atlantic Ocean.

For the autumn PDO mode, significant heat is transported into the Arctic over Alaska, but to the east of it, heat flows out of the Arctic by an anticyclonic circulation. A heat inflow into the Arctic also occurs over Iceland and Greenland. For the autumn PDO mode, an anticyclonic circulation over Northern Europe transports heat flow into and out of the Arctic over the northern Atlantic Ocean and Barents Sea, respectively. Heat is also transported out of Arctic over the Barents and Kara Seas and northeastern Canada. The heat energy transported into the Arctic helps Arctic sea ice melting either directly or through albedo feedback. The latent heat energy influences Arctic sea ice concentration through water vapor feedback. Kinetic and potential energy can change the atmospheric circulation, thus affecting Arctic sea ice concentration.

5) Fig. 4. The sea ice concentration varies from 0 to 10 or 0 to 100%, which is not a random variables, strictly speaking. This is why the changes can be seen only on the first year ice area, rather than the central Arctic (white area). Actually, the sea ice there also experiences significant decline.

The white area in the central Arctic cannot be seen by the satellite sensor (SMMR poleward of 84.6 N and SSMI poleward of 87.6 N). The lack of data in this region can be seen in previous studies, such as Parkinson and Cavalieri (2008).
Parkinson, C. L., and D. J. Cavalieri (2008), Arctic sea ice variability and trends, 1979–2006, J. Geophys. Res., 113,C07003, doi:10.1029/2007JC004558.

6) Fig. 9. The AO mode here is not a typical AO mode, it is more like a combined AO and other modes not investigated here.

The reviewer is correct that the AO mode is not typical. We found that there is not a significant trend in summertime AO. We replaced the word "corresponding" by "bear some resemblance"

Based on my professional training and knowledge, I would not recommend it to be published in the Cryosphere, because of above concerns and confusions.

We hope that our answers and new analyses have addressed the reviewer's concerns

Minors:

English needs polishing throughout

English is polished

Fig. 10. The anomalous wind fields are difficult to digest.

We make the regression of zonal and meridional components onto the PC1, respectively, then plot the vector using their regression coefficients. This is now clarified.

Fig. 3. Are the two modes separated?

Using the method of North et al. (1982), 27.8-19.6=8.2>6.3= $27.8 \times \left( \dfrac{2}{39} \right)^{0.5}$ the two

modes are separated.